# Assessing the Relationship between proAKAP4 Level and Longevity of Sexed Sperm Quality after Thawing

**DOI:** 10.3390/vetsci11090444

**Published:** 2024-09-21

**Authors:** İlktan Bastan, Fırat Korkmaz, Derya Şahin, Seher Şimşek, Ufuk Kaya

**Affiliations:** 1Department of Reproduction and Artificial Insemination, Faculty of Veterinary Medicine, Burdur Mehmet Akif Ersoy University,15000 Burdur, Türkiye; fkorkmaz@mehmetakif.edu.tr; 2Department of Biotechnology, International Center for Livestock Research and Training, 06852 Ankara, Türkiye; deryasahin@tarimorman.gov.tr; 3Department of Animal Health and Quarantine, General Directorate of Food and Control, 06800 Ankara, Türkiye; seher.yirtici@tarimorman.gov.tr; 4Department of Biostatistics, Faculty of Veterinary Medicine, Hatay Mustafa Kemal University, 31060 Hatay, Türkiye; u.kaya@mku.edu.tr

**Keywords:** advanced sperm analysis techniques, sperm viability, precursor A-kinase anchor protein 4 biomarker, CASA, flow cytometer, proteomics, Holstein

## Abstract

**Simple Summary:**

The process of acquiring sexed sperm involves the separation of sperm with X and Y chromosomes, allowing for the production of offspring of the desired sex. However, this process can negatively affect the quality of semen due to biochemical and physical stresses. ProAKAP4 is a major component of the sperm flagellum structure. In this study, sexed bull semen samples were evaluated using the proAKAP4 biomarker for the first time. Based on CASA and flow cytometry analysis of spermatozoa after thawing and incubation for 3 h, proAKAP4 is considered a practical biomarker for estimating the longevity of sexed sperm quality.

**Abstract:**

ProAKAP4 is a sperm structural protein that regulates motility through the PKA-dependent cAMP signaling pathway, which is synthesized as an X chromosome-linked member of the gene family. This study aims to determine the optimal level of proAKAP4 for evaluating sexed semen through investigating its relationship with the longevity of sperm quality in sexed Holstein bull sperm. A total of 30 sexed sperm samples (bearing X chromosomes) from 30 distinct Holstein bulls (*n* = 30) were analyzed. The frozen bull sperm samples were assessed for their proAKAP4 levels, mitochondrial membrane potential, plasma membrane and acrosome integrity (PMAI), and spermatozoa movement parameters at hours 0 and 3 after thawing. The proAKAP4 levels in the sexed sperm samples ranged from 16.35 to 72.10 ng/10 M spz, with an average of 37.18 ± 15.1 ng/10 M spz. A strong positive correlation was observed between proAKAP4 levels and total motility, progressive motility, PMAI, high mitochondrial membrane potential, VAP, and VCL values after 3 h of incubation, when compared to post-thaw analyses. The results also reveal that spermatozoa with proAKAP4 levels of ≥40 ng/10 M spz exhibit higher quality. In conclusion, the level of proAKAP4 in sexed sperm aligns with previous studies and shows potential as a biomarker for assessing the longevity of sexed sperm quality.

## 1. Introduction

The use of sexed semen in dairy cattle farming is crucial for acquiring heifers with high genetic potential and ensuring herd continuity [1]. In the production of sexed sperm, the sperm undergo a series of processes. First, sperm are stained with Hoechst 33342—a DNA-binding fluorescent dye—to differentiate between X and Y chromosomes based on DNA content. This staining process takes approximately 1 h. The stained sperm cells are then introduced into a flow cytometer under pressure within a fluid mixture. As these cells pass through short-wavelength laser light in the tube, the X chromosome—which contains more DNA—appears brighter than the Y chromosome. This is because spermatozoa carrying the X chromosome absorb the fluorescent dye more effectively due to their higher DNA content, resulting in approximately 4% more fluorescence compared to Y chromosome-bearing spermatozoa. This fluorescence emission is analyzed using a powerful computer. The sperm pass through the flow cytometer at a speed of approximately 80 km/h in the form of droplets and are charged either positively or negatively, depending on the chromosome they carry (X chromosome being positive). The spermatozoa carrying the X and Y chromosomes are separated using oppositely charged deflection plates [2,3]. However, this process can stress the spermatozoa, causing physical and biochemical damage [4]. Therefore, evaluating the quality of frozen–thawed sexed semen and, more importantly, predicting its quality is essential, as sexed semen must remain viable until ovulation to enable fertilization [5,6]. Sperm quality decreases over time, due to the depletion of sperm energy reserves and increased reactive oxygen species (ROS) levels. This situation is more commonly observed in sexed sperm than conventional sperm [6,7]. Various computer-assisted semen analysis (CASA) and flow cytometry analysis methods are utilized in andrology laboratories to predict the quality potential of sexed sperm. These advanced sperm analysis techniques thoroughly assess parameters such as sperm motility, viability, acrosome integrity, and mitochondrial membrane potential (MMP) in large sperm populations. These analyses are repeated after different incubation periods to predict changes in sperm quality occurring over time [8,9]. However, this approach imposes additional demands in terms of time, workload, and cost in the assessment of sexed semen. In this regard, recent sperm proteomic studies have provided promising results for predicting the longevity of sperm quality [10]. It is estimated that thousands of proteins are present in semen. The main components isolated from these protein molecules which have been found to be related to sperm quality include BSP, ODF2, ACOT9, a-kinase binding protein (AKAP) 3 and 4, precursor AKAP4 (proAKAP4), ATP-5O, SPADH2, TEKT-1, GAPDHS, aSFP, FMP, HBP, HSP, phospholipases, and interleukins [11,12,13,14,15]. The majority of these proteins are derived from the seminal plasma, the composition of which is susceptible to alterations caused by external factors, including individual characteristics, nutritional intake, and the frequency of semen collection [16,17].

AKAP4, a major component (~50%) of the fibrous sheath of the spermatozoon flagellum, is involved in sperm motility, chemotaxis, and capacitation, playing an important role in the fertilization process [18,19]. ProAKAP4 is a precursor protein in the synthesis of the AKAP4 protein. Recent molecular studies have shown a positive correlation between AKAP4 and proAKAP4 levels in mammalian spermatozoa [20,21,22]. AKAP4 and ProAKAP4 are structural proteins that are densely located in the fibrous sheath of the flagellum and which are involved in the regulation of motility through the PKA-dependent cAMP signaling pathway [23,24]. ProAKAP4 is a sperm structural protein which is distinguishable from many other proteins associated with sperm movement, as it can be conveniently detected using sandwich enzyme-linked immunosorbent assay (ELISA), alongside the Western blotting and sodium dodecyl sulfate polyacrylamide gel electrophoresis techniques [20,22].

Significant correlations between proAKAP4 levels and total and progressive motility have been identified in human, bull, stallion, camel, ram, and dog semen, suggesting that proAKAP4 represents the second major component of the fibrous sheath of the spermatozoon flagellum—following AKAP4—and may, therefore, serve as a biomarker protein [20,25,26,27,28,29]. Nevertheless, the level of proAKAP4 required for optimal sperm quality remains to be determined. This variability currently precludes proAKAP4 from functioning as a reliable biomarker for estimating sperm motility or quality. A similar situation has been observed in studies examining the relationship between proAKAP4 and non-sexed semen quality in Simmental, Holstein, and Nelore bulls [26,30,31].

To date, no studies have examined the relationship between proAKAP4 levels and the longevity of sexed sperm quality. As proAKAP4 is an X chromosome-linked member of the gene family, its concentration may differ in sexed sperm [32,33]. Therefore, this study involved an investigation of the relationship between quality and proAKAP4 levels in sexed sperm from Holstein bulls, using CASA and flow cytometry to evaluate sperm motility and kinematic parameters, MMP, plasma membrane, and acrosome integrity at 0 and 3 h after thawing. Thus, the study aimed to determine the optimal concentration of proAKAP4 for evaluating sexed sperm.

## 2. Materials and Methods

### 2.1. Sperm Samples

This study used commercial bull sperm doses, including sexed sperm samples (bearing X chromosomes) from 30 distinct Holstein bulls. Prior to analysis, the frozen sperm samples (0.25 mL straws) were thawed at 37 °C for 30 s.

### 2.2. Sperm Motility and Kinetic Parameters

Sperm motility characteristics were evaluated using the IVOS I CASA system (Hamilton Thorne Inc., Beverly, OR, USA) after thawing (at 37 °C for 30 s) and incubation (at 37 °C for 3 h). The analysis was conducted at 10× magnification. For each sample, 3 µL of semen was placed on a pre-warmed stage (37 °C) using a 20 µm deep Leja 4-chamber slide (Leja, IMV, L’Aigle, France). Five randomly selected fields were analyzed, capturing 30 frames per field at a frame rate of 60 Hz. Spermatozoa with a velocity above 5 µm/s for both average path velocity (VAP) and straight-line velocity (VSL) were evaluated as motile, and those with a velocity above 50 µm/s for VAP and above 70 µm/s for VSL were classified as progressively motile [34].

### 2.3. Flow Cytometry

A flow cytometry analysis was conducted using a Cytoflex Flow Cytometer (Beckman Coulter, Brea, CA, USA) after thawing semen at 37 °C for 30 s and subsequent incubation at the same temperature for 3 h. The samples were examined using a laser beam with a wavelength of 488 nm (an emitted laser power of 50 mW) and filters with wavelengths of 525 nm, 585 nm, and 610 nm for the detection of emitted fluorescence. Data were collected from 10,000 recorded events following appropriate gate selection [35].

#### 2.3.1. Plasma Membrane and Acrosome Integrity (PMAI)

The integrity of the sperm acrosome and plasma membrane was evaluated using a fluorescein isothiocyanate-conjugated peanut agglutinin (FITC-PNA) and propidium iodide (PI) double-staining methodology. A solution comprising 2.5 μL of FITC-PNA (100 μg/mL) and 1.5 μL of PI (2.99 mM) was added to 245 μL of a buffer solution prepared with 10 μL of semen which had been previously diluted in phosphate-buffered saline solution. This resulted in a final concentration of 5 × 10^6^ sperm/mL. All samples were incubated for 15 min at 37 °C. The samples were then analyzed using the CytExpert 2.2 software (Beckman Coulter) [35].

#### 2.3.2. Mitochondrial Membrane Potential

MMP was assessed using molecular probes, namely, 5,5′,6,6′-tetrachloro-1,1′3,3′-tetramethyl benzimidazolyl-carbocyanine iodide (JC-1) and PI. One straw was thawed (at 37 °C for 30 s), and a spermatozoa suspension (10 µL, containing 5 × 10^6^ spermatozoa) was added to 487 µL of PBS. Then, 10 µL of JC-1 (T3198, 0.153 mM) and 3 µL of PI (L7011, 2.99 mM) were added, and the samples were incubated for 30 min at 37 °C in a dark room, using a water bath to maintain the required temperature within the experimental setup. After incubation, debris was excluded and high MMPs (HMMPs) were analyzed using the CytExpert 2.2 software [36].

### 2.4. ProAKAP4 Assay with ELISA

The ProAKAP4 levels (ng/10 million spermatozoa) in sexed semen samples were determined via ELISA using a Bull 4MID Kit (4BioDx, 4VDX-18K4, Lille, France), according to the manufacturer’s instructions. After thawing, 25 µL of semen was combined with 125 µL of a commercial lysis buffer for ELISA quantification. Then, 100 µL of the samples was loaded onto a 96-well microplate coated with the proAKAP4 anti-body. The mixture was incubated for 90 min. After washing, 100 µL of proAKAP4 detection antibodies was added to each well, allowing for the covalent bonding of horseradish peroxidase for 30 min. Subsequently, the substrate solution was introduced to each well, resulting in a color reaction proportional to the quantity of proAKAP4 in the sperm samples. The reaction was terminated with stop solution, and the color intensity was measured via spectrophotometry at 450 nm (Biotek Instruments, 800TS Microplate Reader, Winooski, VT, USA) [30].

### 2.5. Statistical Analysis

Data analysis was conducted using Stata 12/MP4 statistical software (StataCorp LP, College Station, TX, USA). A series of descriptive statistical analyses were conducted on the data, with the resulting values presented as mean ± standard error values. The normality of the data distribution was analyzed using the Shapiro–Wilk method. The Levene test was employed to analyze the homogeneity of variances between the groups. The influence of groups, time, and the interaction between group and time on the relevant variables was examined through mixed model analysis. In this model, time, group, and the group-by-time interaction were treated as fixed effects, while individual animals were considered random effects. When the interaction term was found to be significant, a simple effect analysis with Bonferroni correction was performed. Pearson’s correlation coefficient was used to assess the relationship between spermatological parameters and proAKAP4 levels. Normally distributed data were analyzed using one-way analysis of variance, and non-normally distributed data were examined with the Kruskal–Wallis test. For all statistical tests, significance was set at a *p*-value of less than 0.05.

## 3. Results

### 3.1. ProAKAP4 Levels of Sexed Semen Samples

The proAKAP4 levels in sexed semen samples ranged between 16.35 ng/10 M spz and 72.10 ng/10 M spz, with an average of 37.18 ± 15.1 ng/10 M spz. The sexed semen samples were categorized into three groups, according to their proAKAP4 concentration: high concentration (HC) with ≥40 ng/10 M spz (*n* = 12), moderate concentration (MC) with 25 ng/10^6^ to 39 ng/10 M spz (*n* = 9), and low concentration (LC) with <25 ng/10 M spz (*n* = 9).

### 3.2. Evaluation of Sperm Motility, PMAI, and HMMP after Thawing and Incubation for 3 h

Table 1 presents the results for total motility, progressive motility, PMAI, and HMMP after thawing and after 3 h of incubation. There were no statistically significant differences between the three groups in terms of the total motility (TM_0_) and progressive motility (PM_0_) values measured immediately after thawing. However, significant differences were observed in PMAI and HMMP, with the highest values for both found in the HC group, in post-thawed sexed semen (*p* < 0.001). After 3 h of incubation, statistically significant differences were observed in total motility (TM_3_), PMAI, and HMMP sperm values among the three groups, with the highest values found in the HC group and the lowest in the LC group (*p* < 0.001). Furthermore, the overall mean value of PM was found to be statistically different after the incubation period (*p* < 0.001).

### 3.3. Sperm Kinetic Parameters of the proAKAP4 Groups

The time, group, and group-by-time interaction values for the sperm kinetic parameter variables are shown in Table 2. In sperm kinematic parameters following thawing, no statistically significant differences were observed between the groups, except for linearity (LIN). The time, group, and group-by-time interaction statistically significantly differed between the groups for the curvilinear velocity (VCL) value (*p* < 0.05). Although the VCL value measured at Hour 0 after thawing was similar between the groups, it was significantly lower in the LC group (91.94 ± 2.28 µm/s) after 3 h of incubation (*p* < 0.05). The VSL, straightness (STR), and LIN values after thawing and incubation for 3 h were similar between the groups (*p* > 0.05), but the effect of time on these values was significant (*p* < 0.05). The statistical evaluations of the VAP, amplitude of lateral head displacement (ALH), and beat-cross frequency (BCF) values were significant for the group and time interactions (*p* < 0.05) and non-significant for the group × time interaction (*p* > 0.05).

### 3.4. Correlation between proAKAP4 and Sperm Parameters

In post-thawed semen, proAKAP4 levels correlated positively with PMAI (r_0_ = 0.572, *p* < 0.01), PM (r_0_ = 0.374, *p* < 0.05), and HMMP (r_0_ = 0.405, *p* < 0.05), while they were negatively correlated with LIN (r_0_ = −0.420, *p* < 0.05) (Figure 1B–D). After 3 h of incubation, a strong positive correlation was found between proAKAP4 levels and TM (r_3_ = 0.878, *p* < 0.001), PM (r_3_ = 0.755, *p* < 0.001), PMAI (r_3_ = 0.744, *p* < 0.001), HMMP (r_3_ = 0.894, *p* < 0.001), VAP (r_3_ = 0.607, *p* < 0.001), and VCL (r_3_ = 0.788, *p* < 0.001) (Figure 1).

## 4. Discussion

The sex-sorting process imposes significant biochemical and physical stresses on sexed sperm when compared to non-sexed sperm [37]. Additionally, as sexed sperm are largely separated from seminal plasma, they may exhibit earlier capacitation and hyperactivity than non-sexed sperm [38,39]. Recent proteomics studies have shown lower levels of certain acrosomal, cytoskeletal, and extracellular proteins in sexed sperm, when compared to non-sexed sperm [33]. Consequently, it is generally believed that sexed sperm, with a shortened fertile lifespan, have lower artificial insemination success [40]. Some researchers have suggested that, due to the potentially shorter lifespan, artificial insemination with sexed sperm should be performed at the time of ovulation (rather than during estrus) in order to increase the success rate [41,42].

The spermatological values obtained in the current study support the notion that the lifespan of sexed sperm is short [37,40]. Notably, after the 3 h incubation period, the TM_3_, PM_3_, PMAI_3_, and HMMP_3_ values in the MC and LC groups were unacceptably low. The intense energy required for stable movement of spermatozoa is supplied by adenosine triphosphate (ATP) through glycolysis and oxidative phosphorylation [43]. The restructuring of the plasma membrane increases intracellular calcium levels. This process stimulates tyrosine phosphorylation through binding of the phosphate group (released during ATP hydrolysis) to AKAP4 and proAKAP4 via PKA enzymes, leading to an increase in intracellular cAMP levels. This, in turn, enhances glycolysis and ATP production [44,45,46]. The low TM, PM, PMAI, and HMMP values observed in the MC and LC proAKAP4 groups are considered to be potentially related to this mechanism.

Almeida et al. [25] reported that the VCL value was significantly higher than the VAP and VSL values and, contrary to the current study, VCL was observed to be positively correlated with LIN. A similar observation was made in a study undertaken by Ruelle et al. [31]. This suggests that the sperm used in these two studies exhibited a high degree of circular motion after thawing. This condition is not desirable in post-thaw sperm evaluation, as it indicates thermal shock or early hyperactivation [47,48].

The post-thawed motility values reported in the above-mentioned studies are similar to those observed in the current study. Almeida et al. [25] and Ruelle et al. [31] supported their findings based on proAKAP4 and spermatological analysis results by examining the relationship between the non-return rate at 90 days after insemination and proAKAP4 concentration. In the current study, the non-return rate results could not be shared as the semen analyzed in the study were used for artificial insemination in the field under different field conditions and at different times, and the respective companies deemed it inappropriate to disclose these data. However, based on our preliminary evaluation and the spermatological findings of Almeida et al. [25] and Ruelle et al. [31], it is anticipated that there would also be a positive correlation between proAKAP4 levels and non-return rates in sexed sperm.

In a study conducted by Baştan and Akçay [30] focusing on un-sexed Simmental sperm, similar to the current study, sperm samples with proAKAP4 concentrations of 40 ng/mL or higher exhibited lower losses in TM and PM values after a 3 h incubation period, compared to other groups. Additionally, when examining kinematic values, these sperm showed a more linear motility pattern [47,49]. This is considered to be related to the fact that conventional sperm (i.e., those not subjected to the stress of the sorting process) better maintain their linear motility characteristics after 3 h of incubation.

Sperm kinematic values are crucial for characterizing spermatozoon movement. In this study, the post-thaw proAKAP4 groups had similar VAP and VSL values, with VAP values being significantly higher than VSL values. This indicates excessive lateral head displacement of the spermatozoon, which is further supported by the amplitude of lateral head displacement [46,48]. VCL represents the distance traveled by the spermatozoon along a curvilinear path. While the VCL value was similar among groups immediately after thawing, it was lower in the LC group after 3 h of incubation. The VAP, VSL, and VCL values obtained after the incubation period indicate that spermatozoa in the LC group exhibited more linear motility characteristics. This observation is further supported by the obtained LIN and STR values [48,50]. The BCF value, which indicates the flagellum beat frequency of spermatozoa, was similar across groups, both immediately after thawing and after incubation. However, when evaluated as an overall mean, the BCF value significantly decreased after the 3 h incubation period, indicating a significant depletion of spermatozoon energy reserves over time. The obtained TM, PM, and HMMP values further support the observed BCF_3_ value.

Previous studies have demonstrated a positive correlation between proAKAP4 levels (as determined by means of ELISA and CASA techniques at Hour 0 after thawing) and TM, PM, BCF, LIN, VCL, and VSL in fresh or post-thawed semen samples from stallions, rams, and dogs [20,27,29]. However, contradictory findings have been reported. In studies investigating the correlation between spermatological parameters and proAKAP4 levels in post-thawed bull semen, Bastan and Akcay [30] observed no significant correlation, whereas Dordas-Perpinyà et al. [51] reported positive correlations with TM, PM, and LIN; furthermore, Almeida et al. [25] identified positive correlations with TM, PM, and rapid sperm. Ruelle et al. [31] also observed a correlation between PM, LIN, and BCF. In studies investigating the relationship between proAKAP4 levels and spermatological parameters after 3 h of incubation, Bastan and Akcay [30] reported a positive correlation with TM, PM, VCL, and VSL in bull semen, while Dordas-Perpinyà et al. [52] reported a positive correlation with TM and PM in stallion semen. The majority of these studies have shown that proAKAP4 is correlated with sperm exhibiting proper linear motility, which is characteristic of sperm with high fertilization potential. The present study supports these findings and further reveals a significant correlation between proAKAP4 levels and PMAI and HMMP values after 3 h of incubation. As recent research on the proteomic assessment of sexed semen has indicated, this may be due to the higher abundance of acrosomal and cytoskeletal proteins in non-sexed sperm, while mitochondrial membrane proteins are more abundant in sexed sperm. It is also suggested that this could be related to the associations of these proteins with specific chromosome gene families, including proAKAP4 [22,32,33,53]. Previous studies have reported that low-quality bull semen samples typically have proAKAP4 levels at or below 14, 20, or 25 ng per 10 M spermatozoa [25,30,31,51]. In contrast, high-quality sperm samples consistently show proAKAP4 levels exceeding 40–50 ng per 10 M spermatozoa. Consistent with these findings, our study identified poor-quality sperm as having proAKAP4 levels below 25 ng per 10 M spermatozoa, while higher sperm motility and longevity were observed in samples with proAKAP4 levels of ≥40 ng per 10 M spermatozoa. These results support proAKAP4 as a potential biomarker for assessing the longevity of sexed sperm quality after thawing.

## 5. Conclusions

The fertilization potential of sperm relies on multiple functions, including migration, hyperactivation, and acrosome reaction; therefore, it cannot be assessed through a single analysis. However, proteomic biomarkers such as proAKAP4 can serve as indicators of sperm motility or quality. To establish the optimal proAKAP4 levels for sexed sperm, further research using frozen semen from various breeds and extender processes is necessary, supported by field data.

## Figures and Tables

**Figure 1 vetsci-11-00444-f001:**
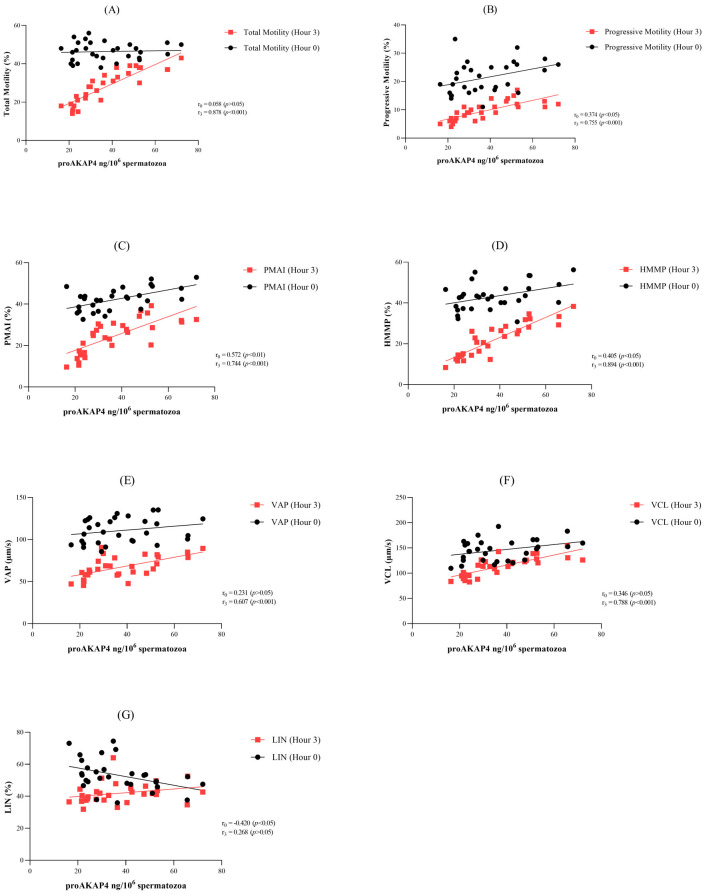
Correlation plots of various sperm motility parameters immediately after thawing and 3 h after thawing: (**A**) total motility, (**B**) progressive motility, (**C**) plasma membrane and acrosome integrity (PMAI), (**D**) high mitochondrial membrane potentials (HMMPs), (**E**) average path velocity (VAP), (**F**) curvilinear velocity (VCL), and (**G**) linearity (LIN).

**Table 1 vetsci-11-00444-t001:** Motility, PMAI, and HMMP of sexed sperm across the proAKAP4 groups.

Parameters	proAKAP4 Groups	Time		Main Effect of Group	*p*		
		Hour 0	Hour 3		Group Effect	Time Effect	Group × Time Interaction
**Total motility** **(%)**	**HC**	46.17 ± 1.00 ^a^	36.50 ± 1.06 ^b, A^	41.33 ± 1.23	<0.001	<0.001	<0.001
	MC	47.78 ± 1.91 ^a^	27.11 ± 1.43 ^b, B^	37.44 ± 2.76			
	LC	45.22 ± 1.77 ^a^	17.67 ± 1.00 ^b, C^	31.44 ± 3.48			
	Overall mean	46.37 ± 0.87	28.03 ± 1.59				
**Progressive motility** **(%)**	HC	23.58 ± 1.43	12.67 ± 0.62	18.13 ± 1.37 ^A^	0.003	<0.001	0.401
	MC	19.78 ± 1.70	8.89 ± 0.56	14.33 ± 1.58 ^B^			
	LC	19.67 ± 2.17	6.11 ± 0.48	12.89 ± 1.97 ^B^			
	Overall mean	21.27 ± 1.03 ^a^	9.57 ± 0.60 ^b^				
**PMAI** **(%)**	HC	45.90 ± 1.34 ^a, A^	31.20 ± 1.48 ^b, A^	38.55 ± 1.82	<0.001	<0.001	<0.001
	MC	39.59 ± 1.38 ^a, B^	26.16 ± 1.20 ^b, B^	32.88 ± 1.85			
	LC	39.83 ± 1.69 ^a, B^	14.47 ± 1.22 ^b, C^	27.15 ± 3.24			
	Overall mean	42.18 ± 0.99	24.67 ± 1.51				
**HMMP** **(%)**	HC	45.21 ± 2.10 ^a, A^	29.83 ± 1.26 ^b, A^	37.52 ± 2.00	<0.001	<0.001	<0.001
	MC	44.03 ± 2.01 ^a, AB^	19.95 ± 1.68 ^b, B^	31.99 ± 3.18			
	LC	39.43 ± 1.65 ^a, B^	12.70 ± 0.69 ^b, C^	26.07 ± 3.36			
	Overall mean	43.12 ± 1.20	21.73 ± 1.51				

HC: high concentration (proAKAP4 ≥ 40 ng/10 million spermatozoa), MC: moderate concentration (25 ng ≤ proAKAP4 < 40 ng/10 million spermatozoa), LC: low concentration (proAKAP4 < 25 ng/10 million spermatozoa), PMAI: plasma membrane and acrosome integrity, HMMP: high mitochondrial membrane potentials. ^a, b^ Statistically significant differences for each parameter are indicated by different letters on the same line (*p* < 0.001). ^A, B, C^ Statistically significant differences for each parameter are indicated by different letters in the same column (*p* < 0.001).

**Table 2 vetsci-11-00444-t002:** Sperm kinetic parameters of the proAKAP4 groups.

	proAKAP4 Groups	Time		Main Effect of Group	*p*		
		Hour 0	Hour 3		Group Effect	Time Effect	Group × Time Interaction
**VAP** **(µm/s)**	**HC**	113.83 ± 4.35	72.51 ± 3.57	93.17 ± 5.11 ^A^	0.018	<0.001	0.200
	MC	109.18 ± 5.33	71.69 ± 3.66	90.43 ± 5.52 ^AB^			
	LC	107.80 ± 4.69	55.22 ± 2.21	81.51 ± 6.85 ^B^			
	Overall mean	110.63 ± 2.70 ^a^	67.08 ± 2.35 ^b^				
**VCL** **(µm/s)**	HC	149.87 ± 5.55 ^a^	128.29 ± 3.22 ^b, A^	139.08 ± 3.86	0.004	<0.001	0.010
	MC	147.66 ± 8.40 ^a^	114.78 ± 5.15 ^b, A^	131.22 ± 6.22			
	LC	138.27 ± 6.46 ^a^	91.94 ± 2.28 ^b, B^	115.11 ± 6.53			
	Overall mean	145.72 ± 3.85	113.33 ± 3.47				
**VSL** **(µm/s)**	HC	71.82 ± 1.83 ^a^	59.52 ± 2.61 ^b^	65.67 ± 2.02	0.825	<0.001	0.008
	MC	79.17 ± 2.36 ^a^	55.42 ± 3.11 ^b^	67.29 ± 3.45			
	LC	77.42 ± 1.85 ^a^	57.46 ± 3.85 ^b^	67.44 ± 3.19			
	Overall mean	75.70 ± 1.27	57.67 ± 1.78				
**ALH** **(µm/s)**	HC	8.33 ± 0.24	6.94 ± 0.19	7.63 ± 0.21	0.838	<0.001	0.394
	MC	8.23 ± 0.32	6.81 ± 0.18	7.52 ± 0.25			
	LC	8.12 ± 0.22	7.22 ± 0.19	7.67 ± 0.18			
	Overall mean	8.24 ± 0.15 ^a^	6.99 ± 0.11 ^b^				
**STR** **(%)**	HC	64.38 ± 3.55 ^a^	55.57 ± 3.79 ^b^	59.97 ± 2.70	0.206	<0.001	0.013
	MC	73.57 ± 3.66 ^a^	62.24 ± 1.53 ^b^	67.91 ± 2.37			
	LC	72.93 ± 3.70 ^a^	53.09 ± 2.39 ^b^	63.01 ± 3.22			
	Overall mean	69.70 ± 2.19	56.83 ± 1.82				
**LIN** **(%)**	HC	48.30 ± 1.41 ^a, B^	43.08 ± 1.45 ^b^	45.69 ± 1.13	0.420	<0.001	0.001
	MC	55.53 ± 4.42 ^a, AB^	44.11 ± 3.08 ^b^	49.82 ± 2.96			
	LC	56.89 ± 2.91 ^a, A^	38.04 ± 1.13 ^b^	47.47 ± 2.74			
	Overall mean	53.05 ± 1.78	41.88 ± 1.20				
**BCF** **(Hz)**	HC	33.06 ± 0.66	28.44 ± 0.53	30.75 ± 0.63 ^A^	<0.001	<0.001	0.161
	MC	28.50 ± 0.65	26.33 ± 0.48	27.42 ± 0.47 ^B^			
	LC	32.27 ± 1.21	27.53 ± 0.71	29.90 ± 0.89 ^A^			
	Overall mean	31.45 ± 0.59 ^a^	27.54 ± 0.36 ^b^				

HC: high concentration (proAKAP4 ≥ 40 ng/10 million spermatozoa), MC: moderate concentration (25 ng ≤ proAKAP4 < 40 ng/10 million spermatozoa), LC: low concentration (proAKAP4 < 25 ng/10 million spermatozoa), VAP: average path velocity, VCL: curvilinear velocity, VSL: straight-line velocity, ALH: amplitude of lateral head displacement, STR: straightness, LIN: linearity, BCF: beat-cross frequency. ^a, b^ Statistically significant differences for each parameter are indicated by different letters on the same line (*p* < 0.001). ^A, B^ Statistically significant differences for each parameter are indicated by different letters in the same column (*p* < 0.001).

## Data Availability

The data will be provided upon request from the authors.

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
