# Peer review of "Assessing the Relationship between proAKAP4 Level and Longevity of Sexed Sperm Quality after Thawing"

_vetsci, 2024, doi:10.3390/vetsci11090444_

Round 1
Reviewer 1 Report
Comments and Suggestions for Authors
The topic is interesting the authors determine the optimal levels of proAKAP4 for evaluating sexed semen 24 by investigating its relationship with the prognostic evaluation of semen viability in sexed Holstein 25 bull semen. Authors are stated the topic well with more interesting Data but some recommendations are needed... Introduction lines42-46 need to be rephrasing,.
Line 68-73 need more references to this part...
Check your references guidelines to the journal
Semen motility need more information and 2more related references on same topic
Please mention the inclusion and exclusion criteria for this topic....
Materials what about informed consent?
What about study limitation?
Line 111-118 there is no references in those two paragraphs please add more related references
Data for table 1and 2 should be separated into 4table
Figure one is very poor ln resolution please add more clear fig...
Discussion is ok but the last paragraph is not needed and need to be reorganized again
Conclusion must be summarized the authors stated (During the fertilization process, sperm must perform multiple functions, such as mi-325 gration, hyperactivation, and acrosome reaction. Therefore, the fertility potential of sperm 326 cannot be determined by a single analysis. However, existing proteomic biomarkers can 327 serve as indicators for the prognostic evaluation of specific sperm functions. In this re-328 spect, proAKAP4 is a potential biomarker for assessing sperm motility or viability. How-329 ever, to determine the optimal proAKAP4 value for optimal sexed sperm motility or via-330 bility, further studies using frozen semen from different breeds and various extender pro-331 cesses are needed. These studies should also be supported by field results.) all those lines should be summarized as clear as possible to be maximum three lines conclude the study
Comments on the Quality of English Language
Need sever English editing
Author Response
Dear Editor,
Thank you very much for supervising the review process of our manuscript (manuscript ID: vetsci-3224940) entitled “Assessing the relationship between proAKAP4 levels and prognostic
viability in sexed semen”. We appreciate the time and effort that you and the reviewers have dedicated to providing your valuable feedback on our manuscript. We are grateful to the reviewers for their insightful comments on our paper. We have been able to incorporate changes to reflect most of the suggestions provided by the reviewers. Accordingly, the revised manuscript has been systematically improved with new information and additional interpretations. We have highlighted the changes on the manuscript.
We request you to accept the responses and the revised manuscript for further process and possible publication in your kind journal. We look forward to hearing from you at your earliest convenience. Thank you in anticipation.
Here is a point-by-point response to the reviewers’ comments and concerns.
Yours Sincerely,
Corresponding author
Comments from Reviewer 1
Comment 1: Introduction lines42-46 need to be rephrasing.
Response 1: Thank you for your precious suggestion. The sentence has been revised as follows (line 42-55): In the production of sexed sperm, the sperm undergo a series of processes. First, sperm are stained with Hoechst 33342—a DNA-binding fluorescent dye—to differentiate between X and Y chromosomes based on DNA content. This staining process takes approximately 1 hour. The stained sperm cells are then introduced into a flow cytometer under pressure within a fluid mixture. As these cells pass through short-wavelength laser light in the tube, the X chromosome—which contains more DNA—appears brighter than the Y chromosome. This is because spermatozoa carrying the X chromosome absorb the fluorescent dye more effectively due to their higher DNA content, resulting in approximately 4% more fluorescence compared to Y chromosome-bearing spermatozoa. This fluorescence emission is analyzed using a powerful computer. The sperm pass through the flow cytometer at a speed of approximately 80 km/h in the form of droplets and are charged either positively or negatively, depending on the chromosome they carry (X chromosome being positive). The spermatozoa carrying the X and Y chromosomes are separated using oppositely charged deflection plates [2,3].
Comments 2: Check your references guidelines to the journal.
Response 2: I have reviewed the journal's reference guidelines and ensured that all citations in the manuscript comply with their requirements.
Comments 3: Semen motility need more information and 2more related references on same topic.
Response 3: The relevant references have been included (line 392-395).
Rahamim Ben-Navi, Liat, Tal Almog, Zhong Yao, Rony Seger, and Zvi Naor. A-Kinase Anchoring Protein 4 (AKAP4) is an ERK1/2 substrate and a switch molecule between cAMP/PKA and PKC/ERK1/2 in human spermatozoa. Sci. Rep., 2016, 6, 37922.
Zhang, K.; Xu, X.H.; Wu, J.; Wang, N.; Li, G.; Hao, G.M.; Cao, J.F. Decreased AKAP4/PKA signaling pathway in high DFI sperm affects sperm capacitation. AJA, 2024, 26(1).
Comments 4: Materials what about informed consent?
Response 4: In the study, the frozen sex-sorted semen of bulls imported into our country was used. Approval documents for research and publication permissions from the institute where the study was conducted are available. Furthermore, the study was financially supported by the institute's research body, which made it possible to carry out the research. I can share these documents if you are requesting them.
Comments 5: What about study limitation?
Response 5: The study limitations are discussed in lines 267–276, as indicated in the following sentence: The post-thawed motility values reported in the above-mentioned studies are similar to those observed in the current study. Almeida et al. [25] and Ruelle et al. [31] supported their findings based on proAKAP4 and spermatological analysis results by examining the relationship between the non-return rate at 90 days after insemination and proAKAP4 concentration. In the current study, the non-return rate results could not be shared as the semen analyzed in the study was used for artificial insemination in the field under dif-ferent field conditions and at different times, and the respective companies deemed it in-appropriate to disclose this data. However, based on our preliminary evaluation and the spermatological findings of Almeida et al. [25] and Ruelle et al. [31], it is anticipated that there would also be a positive correlation between proAKAP4 levels and non-return rates in sexed sperm.
Comments 6: Line 111-118 there is no references in those two paragraphs please add more related references.
Response 6: Thank you for your suggestion. The relevant reference has been included in the text.
Comments 7: Data for table 1and 2 should be separated into 4table
Response 7: Thank you for your precious suggestion. Since these spermatological parameters are closely related to each other and therefore need to be compared, and since they are generally presented in this way in articles, the data are shared in this way.
Comments 8: Figure one is very poor ln resolution please add more clear fig...
Response 8: Thank you for your suggestion. The figure with appropriate specifications has been added.
Comments 9: Discussion is ok but the last paragraph is not needed and need to be reorganized again
Response 9: Thank you for your precious suggestion. The last paragraph has been revised as follows in line with your suggestion: Previous studies have reported that low-quality bull semen samples typically have proAKAP4 levels at or below 14, 20, or 25 ng per 10 M spermatozoa [25,30,31,51]. In con-trast, high-quality sperm samples consistently show proAKAP4 levels exceeding 40–50 ng per 10 M spermatozoa. Consistent with these findings, our study identified poor-quality sperm as having proAKAP4 levels below 25 ng per 10 M spermatozoa, while higher sperm motility and longevity were observed in samples with proAKAP4 levels of ≥ 40 ng per 10 M spermatozoa. These results support proAKAP4 as a potential biomarker for as-sessing the longevity of sexed sperm quality after thawing.
Comments 10: Conclusion must be summarized the authors stated (During the fertilization process, sperm must perform multiple functions, such as mi-325 gration, hyperactivation, and acrosome reaction. Therefore, the fertility potential of sperm 326 cannot be determined by a single analysis. However, existing proteomic biomarkers can 327 serve as indicators for the prognostic evaluation of specific sperm functions. In this re-328 spect, proAKAP4 is a potential biomarker for assessing sperm motility or viability. How-329 ever, to determine the optimal proAKAP4 value for optimal sexed sperm motility or via-330 bility, further studies using frozen semen from different breeds and various extender pro-331 cesses are needed. These studies should also be supported by field results.) all those lines should be summarized as clear as possible to be maximum three lines conclude the study.
Response 10: Thank you for your precious suggestion. Conculusion has been summarized according to your suggestion, as follows: The fertilization potential of sperm relies on multiple functions, including migration, hyperactivation, and acrosome reaction and, therefore, cannot be assessed through a single analysis. However, proteomic biomarkers such as proAKAP4 can serve as indicators of sperm motility or quality. To establish the optimal proAKAP4 levels for sexed sperm, further research using frozen semen from various breeds and extender processes is necessary, supported by field data.
Comments 11: Need sever English editing.
Response 11: The English editing was completed by MDPI English editing service.

Reviewer 2 Report
Comments and Suggestions for Authors
Considering the total text of the manuscript, I believe that the main target of the study was to correlate the ProAKAP4 levels with the quality and the lifespan of bull sexed semen, based on acceptable values of kinematics, motility, plasma membrane integrity, acrosome integrity and mitochondrial membrane potential. A problem is that the use of “viability” in title, abstract, text and conclusion, is confusing. Usually, viability is associated with the percentage of spermatozoa with intact plasma membrane. For that reason, I propose to replace viability with lifespan or quality, or with any respective expression to make clear for the reader the aim of the study e.g. lines 245-246 … the viability lifespan of sexed sperm is short… , must be changed to the lifespan of sexed sperm is short. Please do it at least in the title, abstract and conclusion.
e.g. Title: Assessing the relationship between proAKAP4 levels and expected lifespan of sexed semen after thawing or Assessing the relationship between proAKAP4 levels and quality of sexed semen after thawing
Line 192: change to “Sperm kinematic parameters…”
Lines 195-196: “Significant differences were also found in curvilinear velocity (VCL), VSL, straightness 195 (STR), and LIN values among the sperm kinematic parameters between the groups after 196 incubation (p < 0.001).” However, in Table 2, A and B letters about differences between groups, are presented only for VCL. What about VSL, STR, and LIN?
Lines 222-234. All these sentences describe the sperm sexing processing, but this is not the main issue of the present study. Please delete this paragraph and the respective references 33,34. Begin the discussion as following: The sperm sexing process imposes significant biochemical and physical stress on sexed sperm compared to non-sexed sperm…
Line 256: Begin as following: Almeida et al. [21] reported… Delete “In a study on bull sperm conducted”.
Author Response
Dear Editor,
Thank you very much for supervising the review process of our manuscript (manuscript ID: vetsci-3224940) entitled “Assessing the relationship between proAKAP4 levels and prognostic
viability in sexed semen”. We appreciate the time and effort that you and the reviewers have dedicated to providing your valuable feedback on our manuscript. We are grateful to the reviewers for their insightful comments on our paper. We have been able to incorporate changes to reflect most of the suggestions provided by the reviewers. Accordingly, the revised manuscript has been systematically improved with new information and additional interpretations. We have highlighted the changes on the manuscript.
We request you to accept the responses and the revised manuscript for further process and possible publication in your kind journal. We look forward to hearing from you at your earliest convenience. Thank you in anticipation.
Here is a point-by-point response to the reviewers’ comments and concerns.
Yours Sincerely,
Corresponding author
Comments from Reviewer 2
Comments 1: Considering the total text of the manuscript, I believe that the main target of the study was to correlate the ProAKAP4 levels with the quality and the lifespan of bull sexed semen, based on acceptable values of kinematics, motility, plasma membrane integrity, acrosome integrity and mitochondrial membrane potential. A problem is that the use of “viability” in title, abstract, text and conclusion, is confusing. Usually, viability is associated with the percentage of spermatozoa with intact plasma membrane. For that reason, I propose to replace viability with lifespan or quality, or with any respective expression to make clear for the reader the aim of the study e.g. lines 245-246 … the viability lifespan of sexed sperm is short… , must be changed to the lifespan of sexed sperm is short. Please do it at least in the title, abstract and conclusion.
Response 1: Thank you for your precious suggestion. The relevant sentences have been edited as “quality” or “longevity of sexed sperm quality” and are indicated in yellow in the draft manuscript.
Comments 2: Title: Assessing the relationship between proAKAP4 levels and expected lifespan of sexed semen after thawing or Assessing the relationship between proAKAP4 levels and quality of sexed semen after thawing
Response 2: Thank you so much for your suggestion. The title has been revised as follows:
"Assessing the relationship between proAKAP4 level and longevity of sexed sperm quality after thawing"
Comments 3: Line 192: change to “Sperm kinematic parameters…”
Response 3: Thank you for your precious suggestion. The mentioned comment is corrected in the manuscript.
Comments 4: Lines 195-196: “Significant differences were also found in curvilinear velocity (VCL), VSL, straightness 195 (STR), and LIN values among the sperm kinematic parameters between the groups after 196 incubation (p < 0.001).” However, in Table 2, A and B letters about differences between groups, are presented only for VCL. What about VSL, STR, and LIN?
Response 4: Thank you for your precious suggestion. Edited according to your suggestion. It is detailed in the conclusion section as follows (line 204-215): The time, group, and group-by-time interaction values of the sperm kinetic parameter variables are shown in Table 2. Sperm kinematic parameters following thawing, no statistically significant differences were observed between the groups, except for linearity (LIN). The time, group, and group-by-time interaction statistically significantly differed between the groups for the curvilinear velocity (VCL) value (P<0.05). Although the VCL value measured at hour 0 after thawing was similar between the groups, it was significantly lower in the LC group (91.94 ± 2.28 µm/s) after 3 h incubation (P<0.05). The VSL, straightness (STR), and LIN values after thawing and incubation for 3 hours were similar between the groups (P>0.05), but the effect of time on these values was significant (P<0.05). The statistical evaluation of the VAP, amplitude of lateral head displacement (ALH) and, beat-cross frequency (BCF) values were significant for the group and time (P<0.05) and non-significant for the group x time interaction (P>0.05).
Comments 5: Lines 222-234. All these sentences describe the sperm sexing processing, but this is not the main issue of the present study. Please delete this paragraph and the respective references 33,34. Begin the discussion as following: The sperm sexing process imposes significant biochemical and physical stress on sexed sperm compared to non-sexed sperm…
Response 5: Thank you for your precious suggestion. Edited according to your suggestion. In addition, this paragraph has been added to the introduction in line with the suggestion of the other reviewer (line 42-55).
Comments 6: Line 256: Begin as following: Almeida et al. [21] reported… Delete “In a study on bull sperm conducted”.
Response 6: Thank you for your precious suggestion. The mentioned comment is corrected in the manuscript.

Round 2
Reviewer 1 Report
Comments and Suggestions for Authors
The paper is now improved I accept it
Comments on the Quality of English LanguageGood